# Creating Three-Dimensional Templates of Smiling and Pouting Faces for Different Sex- and Age Groups

**DOI:** 10.3390/jcm11247257

**Published:** 2022-12-07

**Authors:** Hilde Schutte, Marvick S. M. Muradin, Freek Bielevelt, Timen C. ten Harkel, Caroline M. Speksnijder, Antoine J. W. P. Rosenberg

**Affiliations:** 1Department of Maxillofacial Surgery, University Medical Center Utrecht, 3584 CX Utrecht, The Netherlands; 23D Lab Radboudumc, Radboud University Medical Centre, 6525 GA Nijmegen, The Netherlands

**Keywords:** average face, face template, facial expressions, facial animation, smiling, closed smile, pouting, three dimensional, ageing, facial morphology, morphometry, mimic musculature

## Abstract

Smile appearance has a major psychological impact. Orthognathic surgery, which has harmonizing results on skeletal structures, can negatively influence the smile appearance due to soft tissue effects. To enhance the aesthetic effects of orthognathic surgery on soft tissues, reference models for large parts of the hospital’s adherent area are called for. This study aims to create average facial models for different sex and age groups in two facial exercises: maximum closed smile, and pouting. These models were created using coherent point drift and Procrustes algorithms in MATLAB. Principal component analysis was performed, and of 20 surgical landmarks, the in-group variation using standard deviation was calculated. Three distances were analyzed: nasal width, philtral width, and mouth width. To correct for facial size, these distances were analyzed as a ratio of intercanthal width. In total, 328 healthy subjects were included in the study. Subjects were grouped by sex, and in age categories spanning four years each, with an adult group with all ages >16 years. For both smiling and pouting faces, all ratios increased with ageing. These templates and data can benefit facial surgeons, to obtain an expected outcome according to the age of the patient.

## 1. Introduction

Orthognathic surgery, the standard procedure to correct dentofacial deformities, has fairly predictable and often harmonizing results on skeletal structures [1,2,3]. Nevertheless, the soft tissue response, especially of the upper lip, is less predictable [4,5]. For example, in le Fort I osteotomies, adverse nasolabial changes include alar flaring, upturning of the nasal tip, and flattening of the upper lip [6,7,8]. To prevent such undesired soft tissue changes, the (modified) alar cinch suture and V-Y closure, as adjunctive procedures, have been invented [4,9]. However, the precise effect of orthognathic surgery on soft tissues, especially in combination with adjunctive procedures, remains an enigma which calls for unraveling [1,10,11,12].

Of all unwanted side effects after orthognathic surgery, it is the smile that shows the most detrimental results [10]. This is crucial, since the psychological impact of a smile can be major; a lot of judgments are based on smile appearance. People distinguish whether or not a smile is authentic, and rate someone’s trustworthiness by it [13,14,15]. Other personality judgments that are based on smiles are likeability, competence, maturity, affiliation, and dominance [16,17]. Subjects with an open smile are rated as better leaders than subjects with a closed smile [18]. Personality traits such as warmth, calmness, extroversion, and low anxiety are closely related to an attractive smile [19]. In children, poor smile esthetics have a negative influence on social interactions [20].

This raises the question of what a ‘normal’ smile is. It is known that smile aesthetics differ between genders, and alter with ageing [19]. Currently, no templates depicting the average face for different age groups and facial exercises are available. The magnitude of specific facial movements says something about the ability to animate. For that reason, two reproducible extreme positions of the oral soft tissues were chosen as subjects of the study: smiling and pouting. Models of these facial positions in healthy subjects can determine the deviation from the average in patients, and can serve as guidelines to strive for. These data can aid in facial surgery, for example in soft-tissue reconstructive cases, to obtain an expected outcome according to the age of the patient. The objective of this study is to provide three-dimensional (3D) templates of average faces in the aforementioned facial expressions for both genders at different age classes. Since the size of the head differs between patients, measurements will be provided as ratios, to facilitate comparison for each individual patient.

## 2. Materials and Methods

### 2.1. Population

Approval for this prospective study was provided by the local ethics committee (study number 14-652). Between December 2016 and January 2017, 3D images were captured of healthy subjects visiting the University Museum in Utrecht, the Netherlands, with the two pod 3dMD system (3dMDface, 3dMD, Atlanta, GA, USA). Informed consent was obtained from all participants. Study approval was provided by the local ethics committee. Each subject was captured in two different facial poses: with a closed smile and pouting. The 3dMD system was placed in a windowless room used for daily clinical 3D imaging, illuminated with 100% LED lighting. The subjects were grouped into age categories spanning four years each. Due to the small number of inclusions in the 16 to 20 year group (6 females and 1 male), it was decided to combine the older groups into one adult group, aged 16 years and older. This resulted in the following age groups: 4 to 8 years, 8 to 12 years, 12 to 16 years, and 16 years and older. For each age category, gender, and facial expression, a separate average face was created, defining 16 groups in total.

### 2.2. Analyses

For each group, four analyses were performed. First, all faces of a group were combined, to create a template of the average face. Second, principal component analysis (PCA) was performed, analyzing the most common variable in a group. This was depicted as a linear function. Third, of all faces, the absolute distance of the paired landmark between each subject’s original face and the averaged face was measured in millimeters (mm), and standard deviation (SD) was calculated. SDs of each landmark were depicted on the average face, with a green to red color scale corresponding to SDs between 0 and ≥3 mm, respectively. Fourth, eight ratios of the face were analyzed. Distances between facial landmarks were provided as ratios, in order to correct for the size of the head. The mathematical environment MATLAB (MATLAB R2020b, The MathWorks, Inc., Natick, MA, USA) was used to process and analyze images, and to calculate distances and ratios. Further explanation of the applied techniques is provided below.

#### 2.2.1. Analysis 1, Average Face

To create an average face from multiple 3D images, each individual 3D image had to be re-meshed, after which the subject-specific templates could be combined into one average face. For the re-meshing process, the following method was applied, with each step depicted in a flowchart in Figure 1. First, the 3D images were pre-processed by automatically filling holes in the mesh, followed by a subsampling to create a uniform mesh with an average vertex distance of 1.0 mm. Next, the following six anatomical landmarks were manually placed on the uniformly distributed mesh: the left and right pupil, pronasale (pn), left and right cheilion (ch), and pogonion (pg) (step 1). These six landmarks were used in the Procrustes algorithm. This algorithm uses the landmarks to align the 3D image to a general template with facial contours, without scaling (step 2). By doing so, all 3D images had the correct rotation and orientation. For each age group, the general face template was scaled according to the six landmarks, in order to account for the variation in head size between the different age groups. Moreover, the 3D image was cropped based on the outer boundary of the face template. Subjects whose forehead was not visible had to be excluded due to technical matching difficulties (step 3).

After the initial alignment and cropping of the 3D image, the landmark-guided coherent point drift (CPD) algorithm was used for the non-rigid deformation of the general face template towards the 3D image. The manually placed left and right pupil, left and right cheilion, and pronasale were used as the landmarks to guide the CPD (step 4). Subsequently, a ray-cast was applied from the vertices of the face template towards the 3D image, where every vertex from the face template was mapped onto the 3D image. Finally, all 3D images were aligned towards the facial template with the Procrustes algorithm using all vertices, by means of rigid registration (step 5). This resulted in every 3D image having the same position and rotation as the general face template, without scaling the face, in order to preserve the true facial measurements. Additionally, all the processed 3D images had the same number of vertices with the same corresponding mesh, resulting in an aligned, subject-specific template. These subject-specific templates were averaged, to create an average face for every one of the four age groups of both genders, and in both poses.

#### 2.2.2. Analysis 2, PCA

The variations in facial morphology within a certain group were analyzed with the principal component analysis (PCA). PCA analyzes the variation within a group, with the first PC being the most common variable. This variable is depicted as a linear function, with the average at ‘0’, and both extremes at ‘−2’ and ‘+2’. For example, when the most common variation from the average is the length of the face, the shortest and longest faces are shown at −2 and +2, with the average face in between, at 0. More information about the technical aspects of PCA can be found online [21].

#### 2.2.3. Analysis 3, SD Analysis

A total of 13 (paired) facial landmarks, with abbreviations listed in Table 1, were analyzed. For each average face, the SD was calculated for each landmark, showing the distance of the paired landmarks between each subject’s original face and the averaged face. Each landmark was depicted on the average face with a green to red color scale corresponding to SDs between 0 and ≥3 mm, respectively.

#### 2.2.4. Analysis 4, Ratio Analysis

To facilitate the comparison for each individual patient, measurements were analyzed as ratios. Three ratios of facial measurements of each individual face were analyzed. Distances were compared to the intercanthal width (ICW: left to right en). Ratio 1: nose width (left to right al); ratio 2: philtrum width (left to right cph); ratio 3: mouth width (left to right ch). The ICW was chosen as referential distance, since the maturation of this distance is reached at 8 years in females, and at 11 years in males, with 44% of the absolute total growth increment occurring between 3 and 4 years of age [22].

### 2.3. Statistical Analysis

Statistical analysis was performed using GraphPad Prism version 8.3.0 for Windows, GraphPad Software, San Diego, CA, USA, www.graphpad.com (accessed on 7 February 2022). Normality was tested using Q-Q plots, and showed normally distributed data. Therefore, data were expressed by means and SDs. Differences between age groups within each gender were analyzed using one-way ANOVA analysis of variants. For statistically significant differences, multiple comparison analyses were performed between all groups, using Tukey’s post hoc test. Statistically significant difference was considered at *p*-values < 0.05.

## 3. Results

### 3.1. Baseline Characteristics

In total, 406 healthy subjects were captured. In 78 subjects, the forehead was not visible, leading to exclusion of their data due to technical matching difficulties. The remaining 328 subjects were divided by gender and age, and their baseline characteristics are demonstrated in Table 2.

### 3.2. Closed Smile

#### 3.2.1. Average Faces, First Principal Component (PC), and Group Variation

Average faces of each group and their first PCs for females captured in the closed smile are shown in Figure 2a, and for males in Figure 2b. For both, males and females, the first PC comprised the height–width ratio of the face in all groups. Additionally, the PC of both genders in the age group 4 to 8 years showed the extent of elevation or depression of the corners of the mouth.

SD analysis of the chosen landmarks demonstrated SDs to be the largest for left and right cheilion, pronasale, and pogonion. All other landmarks had SDs below 2 mm.

#### 3.2.2. Analysis of Ratios

Results of the analysis of the ratios are provided in Table 3. Results of the statistical tests are provided in Table 4. For each measurement, graphs were created depicting the ratio for each sex across the age groups (Figure 3, Figure 4 and Figure 5). Additionally, the measurements of the ICW are provided in the tables, and as a graph (Figure 6). All ratios, as well as the ICW, showed a significant increase with ageing.

### 3.3. Pouting

#### 3.3.1. Average Faces, First Principal Component (PC), and Group Variation

Average faces of each group and their first PCs for females captured in the pouting face are shown in Figure 7a, and for males in Figure 7b. For both, males and females, the first PC comprised the height–width ratio of the face, and the extent of pouting in all groups.

SD analysis of landmarks showed highest variation in peri-oral landmarks, especially left and right cheilion, stomion, and labiale inferius. Left and right crista philtri and labiale superius had moderately high SDs, as well as pronasale. Other landmarks had relatively low SDs.

#### 3.3.2. Analysis of Ratios

Results of the analysis of the ratios are provided in Table 5. Results of the statistical tests are provided in Table 6. For each measurement, graphs are created depicting the ratio for each sex across the age groups (Figure 8, Figure 9 and Figure 10). Additionally, the measurements of the ICW are provided in the tables, and as a graph (Figure 11). For women, there were no significant changes in nasal width: ICW (ratio 1, Figure 8). Women’s philtral width: ICW (ratio 2, Figure 9) showed a significant increase between the age groups 4 to 8 years and 12 to 16 years, and a significant decrease between the age groups 12 to 16 years and 16 years and older. Mouth width: ICW (ratio 3, Figure 10) followed the same trend. However, the decrease in the age group 16 years and older was not significant. ICW significantly increased with higher age (Figure 11). For men, all ratios showed a significant increase at higher ages, and also ICW significantly increased with higher age (Figure 8, Figure 9, Figure 10 and Figure 11).

## 4. Discussion

The present study aimed to provide templates of average faces in facial expressions, and to provide ratios of measurements for two facial expressions, closed smile and pouting. We found a significant increase with aging for nasal width, philtral width, and mouth width, for smiling faces in both genders. For pouting faces, men had a significant increase with aging for all three ratios, and women for both the philtral and mouth width. Reconstructions of average faces for different age groups in two facial expressions were made. The first principal component predominantly involved the height–width ratio of the faces. In the youngest age group, a large variability was seen within the appearance for both facial expressions. This might implicate the inability of children towards facial animation compared to adults. However, it is more plausible that the larger variability in young children is a consequence of the incomprehension of the specific facial exercise, due to their young age.

Variance in peri-ocular landmarks was low, indicating that there was minor in-group variance in this area. However, variance in peri-oral landmarks, especially both cheilion, pogonion, and, for pouting faces, the mid-oral landmarks (labiale superius, inferius, and stomion), was high. This variance can be expected, as the subjects were instructed to perform several facial exercises which involved the mouth. The assumption is that the high variance for pouting faces is due to difficulties in executing this movement. Moreover, the forward movement of the lips in pouting probably resulted in high scores for SDs in the medial peri-oral landmarks. Furthermore, it must be noted that variance can also be explained since soft-tissue landmarks were studied, which are not as solid as bony landmarks.

Analysis of ratios showed that (compared to ICW) mouth width, nasal width, and philtral width increased with age for smiling faces. Unexpectedly, ICW increased at older ages as well. ICW was chosen as referential distance, since there should be barely any growth after early childhood [22]. However, a review by Sforza et al. described a significant effect of age on the ICW, and large differences between ethnicities [23]. Whether this last finding would have implications for the studied expressions in the present study remains unclear.

Still, even when corrected for ICW, measurements in the smiling faces increased with age. In males, this increase for all measurements was seen in the pouting faces as well. For females, when pouting, the width of the nose did not change significantly between the age groups. Interestingly, the philtral width increased until the age of 16, but decreased for the oldest age group. It should be mentioned that, due to the scale of the charts, this difference seems major. In fact, it comprises a difference of 2.84% in the intercanthal width. Still, the decrease is significant. This might be explained by the wide age range within this group, since all subjects above the age of 16 years were combined. Our assumption is that the width of the philtrum might decrease at older ages, due to sagging of the skin [24]. Future research could aim to include more subjects in groups with a smaller age range, especially for the >16 years group. This might provide more insight into the age-dependent changes, as found in the present study.

In summary, for both males and females, the nose, mouth, and philtrum while smiling are wider for older ages. For both older males and females, a wider philtrum and mouth with pouting are seen. In males only, however, a wider nose with pouting is encountered. These age-related changes might be due to facial growth. In a previous study from our center, an increase in ratios was also seen for higher-aged groups. However, compliance, which might be lower at young ages, might have influenced the present results.

To the best of our knowledge, this is the first study that creates templates of average faces in facial expressions, and provides ratios of measurements for two facial expressions, closed smile and pouting. A limitation of the study is the inevitable risk of inaccuracy acquired by the technical aspects. In all steps, a small error can occur. For all images, the six landmarks for the Procrustes algorithm were placed manually. Manual placement of landmarks is accompanied by a small error that is clinically accepted [25,26,27]. The remaining landmarks were automatically placed by CPD, which is proven to be deterministic [28], meaning that there is no inter-observer error. It could be stated that placement of landmarks should be checked manually, to minimize inaccuracy. However, as stated before, manual placement of landmarks comprises an error. Moreover, the clinical applicability would substantially decrease if landmark placement should be carried out manually for large numbers of photographs. In the present study, effort was taken to minimize measurement errors. However, it should be taken into account that all errors were added up, resulting in an inaccuracy of the final results. Therefore, SD analysis of each average face does not merely show group variation, but also includes the error of the analysis itself. This can be seen when examining the results of the ICW measurements: The ICWs of the pouting faces are not exactly the same as the ICWs of the smiling faces.

There is a difference between the type of facial expressions in the present study and most facial grading systems [29,30,31]. Most facial grading systems use the open maximum smile. For the present study, however, the maximum closed smile and pouting, both extreme positions of the mouth, were chosen as referential facial expressions. The main objective of this study was to provide reproducible reference material, which gives an indication of perioral muscular activity. Several studies concluded that the maximum closed smile (posed smile) and pouting are the most reproducible facial exercises [32,33,34]. Consequently, the open maximum smile, although it often has a wider range of motion, was not used in the present study.

Discussing the results of the present study, dealing with the “average Dutch face”, it should be stressed that no specific criteria were used during inclusion of the subjects. After collection of data, subjects were grouped in age categories spanning four years. Retrospectively, this was the ideal group size with enough subjects for analysis power, without combining too heterogenic subjects in the same group. However, there was an unbalanced distribution between the age groups. For example, the male group aged 16–20 years consisted of only one subject. It was, therefore, decided to combine all subjects aged >16 years into one group. As mentioned earlier, future research could aim to include more subjects in groups with a smaller age range.

In the present study, templates are provided of average faces, in smiling and pouting positions. Moreover, measurements of ratios were analyzed, to serve as reference material for the clinician. These data can aid in facial surgery, for example in soft-tissue reconstructive cases, to obtain an expected outcome according to the age of the patient. However, when analyzing facial expression, not only the static result should be evaluated. Dynamic analysis of the facial parts can be a valuable addition when examining facial expressions. Future research might aim to analyze the movement of facial landmarks in three dimensions, to create better insights into facial dynamics.

## Figures and Tables

**Figure 1 jcm-11-07257-f001:**
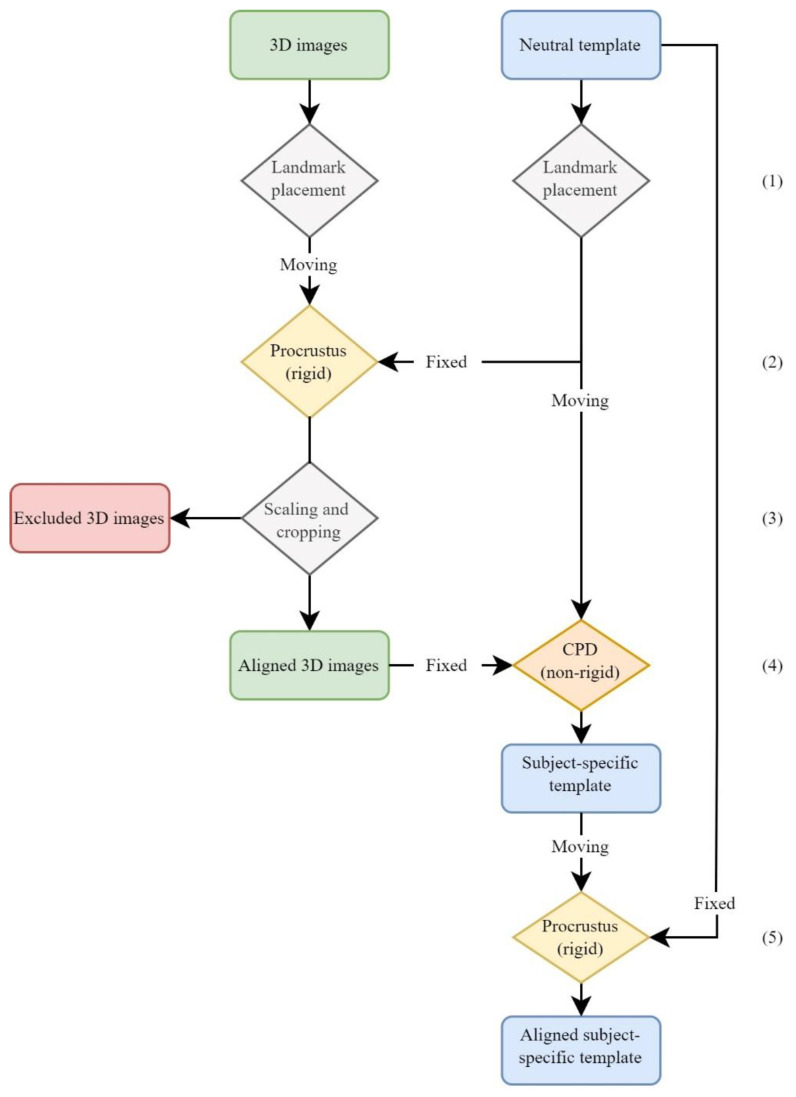
Flowchart of re-meshing process. Summary of all steps that were executed to process the original 3D images into a subject-specific template. These subject-specific templates can be averaged into an average face. CPD: Coherent Point Drift.

**Figure 2 jcm-11-07257-f002:**
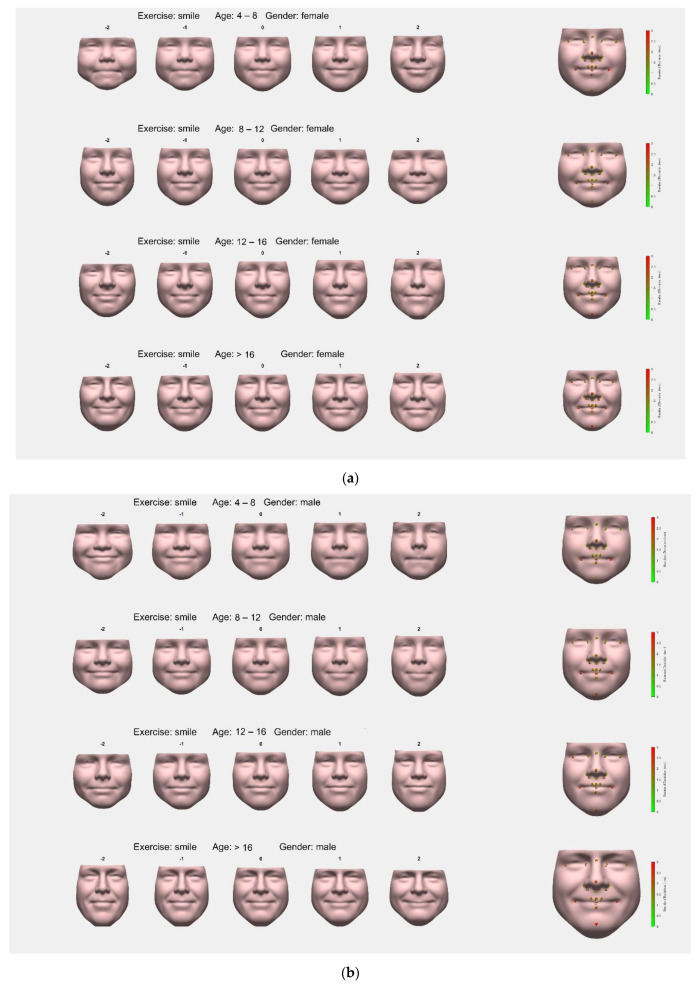
(**a**): Females, closed smile. Average faces for women in the different age classes with a closed smile. The first Principal Component (PC) for each face is provided, depicting the largest variable of each group as a linear function (−2 to +2). On the right side of the picture, facial landmarks are pointed out with colored dots depicting the standard deviation of the corresponding landmark in the group. (**b**): Males, closed smile. Average faces for men in the different age classes with a closed smile. The first Principal Component (PC) for each face is provided, depicting the largest variable of each group as a linear function (−2 to +2). On the right side of the picture, facial landmarks are pointed out with colored dots depicting the standard deviation of the corresponding landmark in the group.

**Figure 3 jcm-11-07257-f003:**
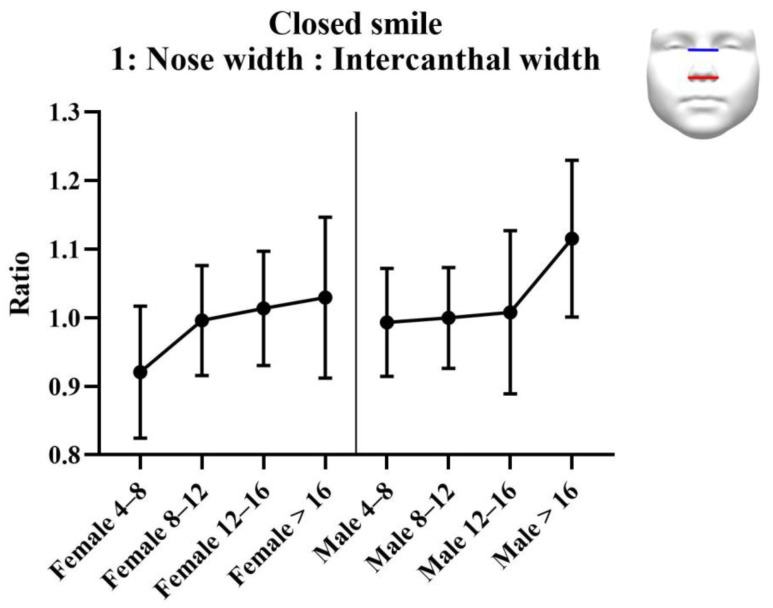
Ratio 1; nose width to intercanthal width ratio for closed smile faces. Nose width is measured from left to right alar. Intercanthal width is measured from left to right endocanthion.

**Figure 4 jcm-11-07257-f004:**
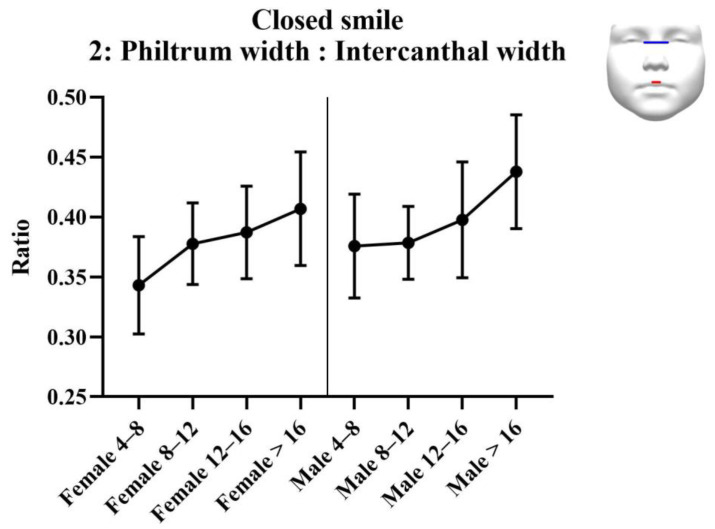
Ratio 2; philtrum width to intercanthal width ratio for closed smile faces. Philtrum width is measured from left to right crista philtri. Intercanthal width is measured from left to right endocanthion.

**Figure 5 jcm-11-07257-f005:**
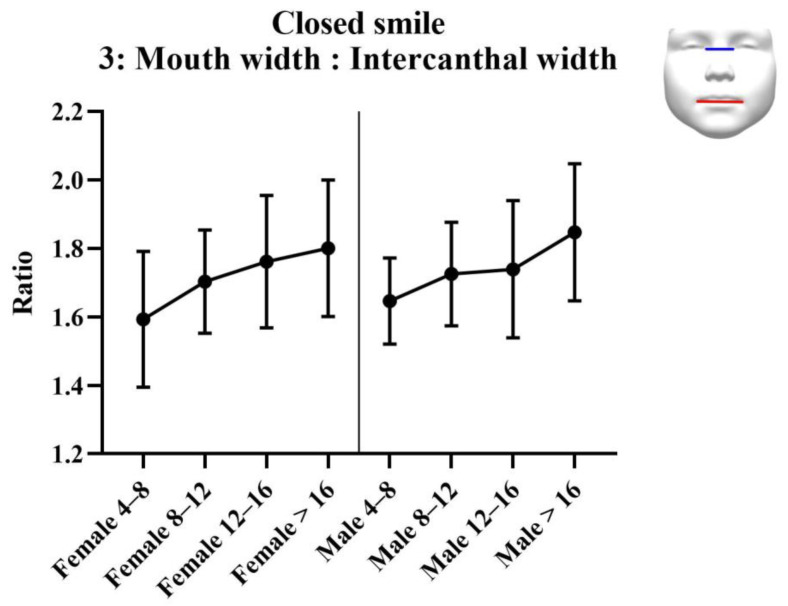
Ratio 3; mouth width to intercanthal width ratio for closed smile faces. Mouth width is measured from left to right cheilion. Intercanthal width is measured from left to right endocanthion.

**Figure 6 jcm-11-07257-f006:**
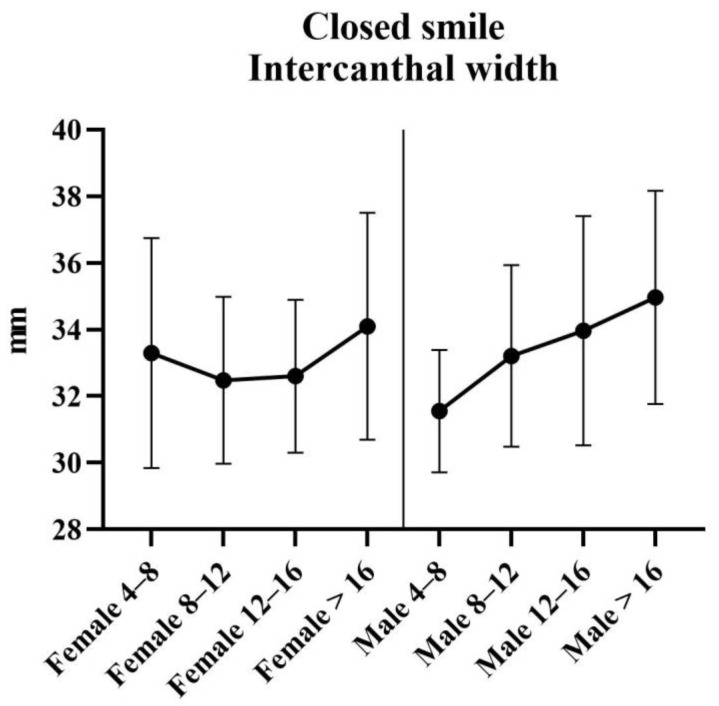
Intercanthal width for closed smile faces. Intercanthal width is measured in millimeters (mm) from left to right endocanthion.

**Figure 7 jcm-11-07257-f007:**
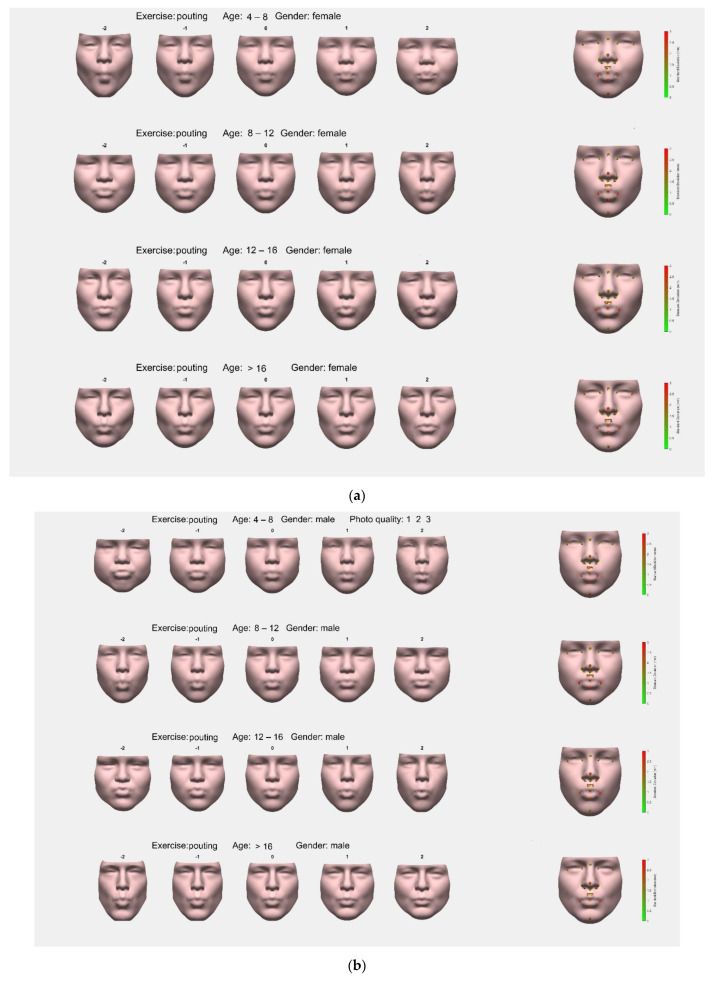
(**a**): Females, pouting. Average faces for women in the different age classes in a pouting position. The first Principal Component (PC) for each face is provided, depicting the largest variable of each group as a linear function (−2 to +2). On the right side of the picture, facial landmarks are pointed out with colored dots depicting the standard deviation of the corresponding landmark in the group. (**b**): Males, pouting. Average faces for men in the different age classes in a pouting position. The first Principal Component (PC) for each face is provided, depicting the largest variable of each group as a linear function (−2 to +2). On the right side of the picture, facial landmarks are pointed out with colored dots depicting the standard deviation of the corresponding landmark in the group.

**Figure 8 jcm-11-07257-f008:**
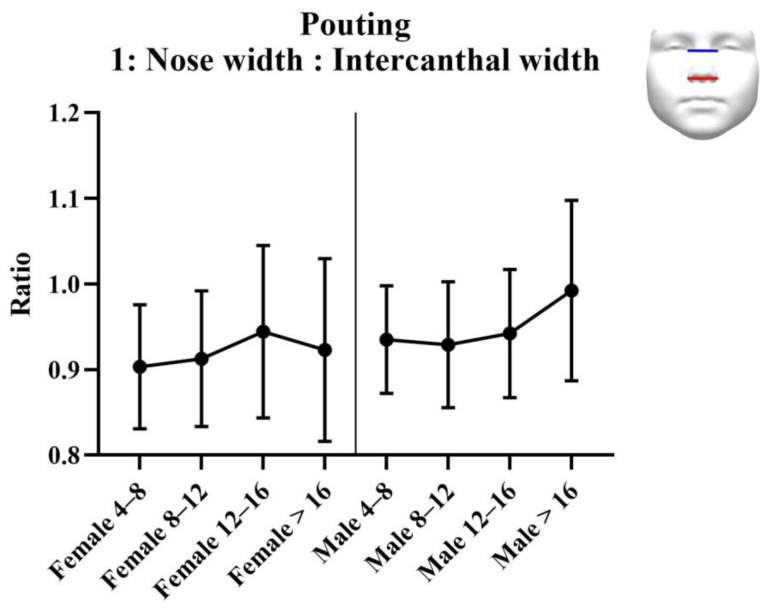
Ratio 1; nose width to intercanthal width ratio for pouting faces. Nose width is measured from left to right alar. Intercanthal width is measured from left to right endocanthion.

**Figure 9 jcm-11-07257-f009:**
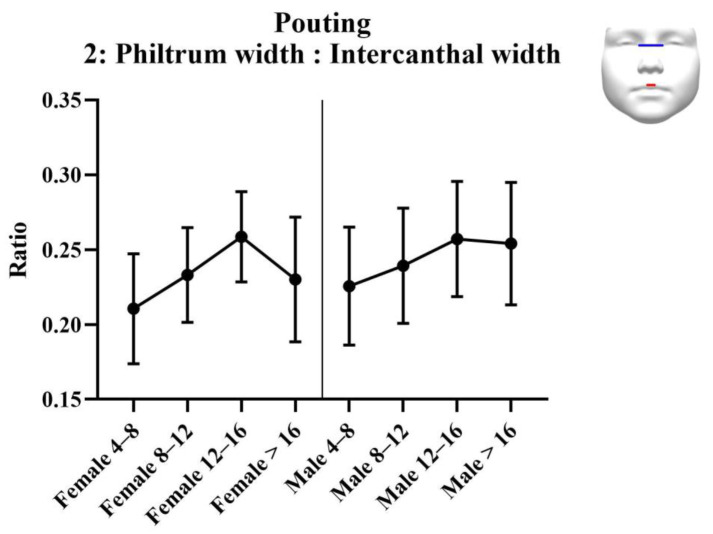
Ratio 2; philtrum width to intercanthal width ratio for pouting faces. Philtrum width is measured from left to right crista philtri. Intercanthal width is measured from left to right endocanthion.

**Figure 10 jcm-11-07257-f010:**
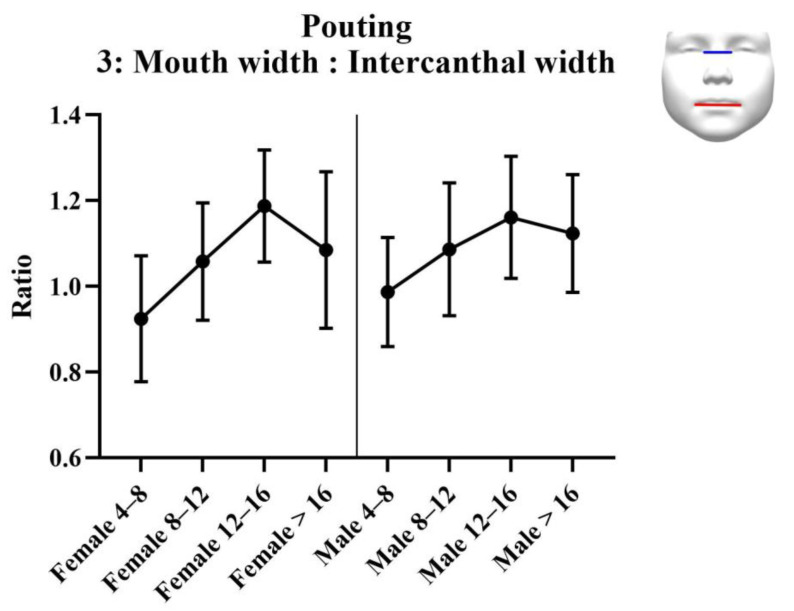
Ratio 3; mouth width to intercanthal width ratio for pouting faces. Mouth width is measured from left to right cheilion. Intercanthal width is measured from left to right endocanthion.

**Figure 11 jcm-11-07257-f011:**
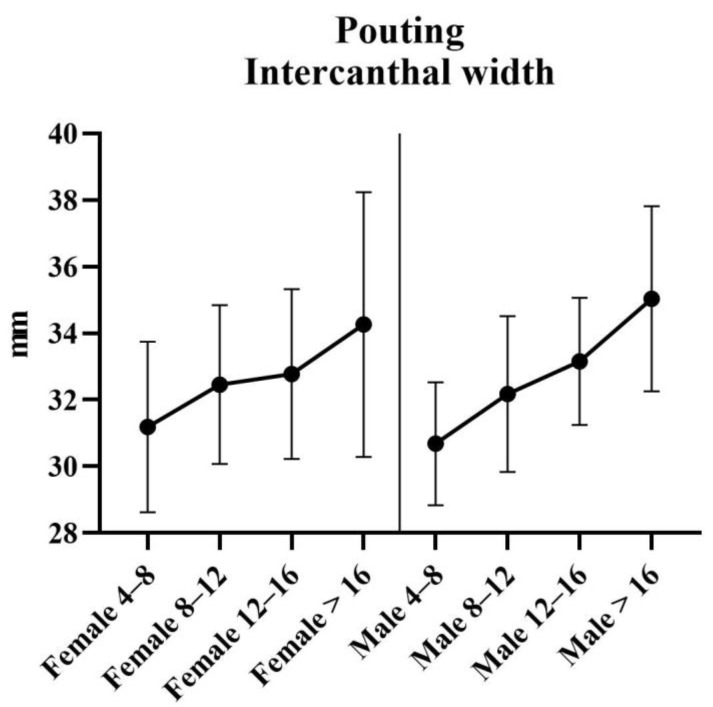
Intercanthal width for pouting faces. Intercanthal width is measured in millimeters (mm) from left to right endocanthion.

**Table 1 jcm-11-07257-t001:** Abbreviations list of facial landmarks.

Al	Alare
ch	Cheilion
cph	Crista philtri
en	Endocanthion
ex	Exocanthion
li	Labiale inferius
ls	Labiale superius
n	Nasion
pg	Pogonion
pn	Pronasale
sbal	Subalare
sn	Subnasale
sto	Stomion

**Table 2 jcm-11-07257-t002:** Baseline characteristics.

Age Group	Female (*n*)	Mean Age * (SD)	Male (*n*)	Mean Age * (SD)
4–8 years	13	6.4 (1.0)	17	6.2 (0.7)
8–12 years	60	9.5 (1.1)	58	9.5 (1.1)
12–16 years	17	12.8 (0.9)	22	12.9 (1.0)
>16 years	78	40.9 (14.0)	63	45.3 (10.4)

All group sizes and their average age in years are provided. * in years. SD: Standard deviation.

**Table 3 jcm-11-07257-t003:** Analysis results: ratios for smiling faces.

Female	Female 4–8 Years (*n* = 13)	Female 8–12 Years (*n* = 60)	Female 12–16 Years (*n* = 17)	Female >16 Years (*n* = 78)
Ratio 1: Nose width: Intercanthal width (SD)	0.92 (0.10)	0.10 (0.08)	1.01 (0.08)	1.03 (0.12)
Ratio 2: Philtrum width: Intercanthal width (SD)	0.34 (0.04)	0.38 (0.03)	0.39 (0.04)	0.41 (0.05)
Ratio 3: Mouth width: Intercanthal width (SD)	1.59 (0.20)	1.70 (0.15)	1.76 (0.19)	1.80 (0.20)
Intercanthal width in mm (SD)	33.3 (3.45)	32.5 (2.51)	32.6 (2.30)	34.1 (3.41)
Male	Male 4–8 years (*n* = 17)	Male 8–12 years (*n* = 58)	Male 12–16 years (*n* = 22)	Male >16 years (*n* = 63)
Ratio 1: Nose width: Intercanthal width (SD)	0.99 (0.08)	1.00 (0.07)	1.01 (0.12)	1.12 (0.11)
Ratio 2: Philtrum width: Intercanthal width (SD)	0.38 (0.04)	0.38 (0.03)	0.40 (0.05)	0.44 (0.05)
Ratio 3: Mouth width: Intercanthal width (SD)	1.65 (0.13)	1.73 (0.15)	1.74 (0.20)	1.85 (0.20)
Intercanthal width in mm (SD)	31.6 (1.84)	33.2 (2.73)	34.0 (3.44)	35.0 (3.20)

SD: Standard deviation; mm: millimeters.

**Table 4 jcm-11-07257-t004:** Analysis results: Statistical analysis for age groups for smiling faces.

Female	Ratio 1 Nose: ICW	Ratio 2 Phi: ICW	Ratio 3 Mouth: ICW	ICW
ANOVA	0.0035 *	<0.0001 *	0.0003 *	0.0147 *
Tukey’s				
Female 4–8 vs. 8–12	0.0739	0.0367 *	0.2034	0.8109
Female 4–8 vs. 12–16	0.0630	0.0240 *	0.0640	0.9245
Female 4–8 vs. >16	0.0024 *	<0.0001 *	0.0012 *	0.8123
Female 8–12 vs. 12–16	0.9181	0.8428	0.6558	0.9987
Female 8–12 vs. >16	0.2208	0.0004 *	0.0118 *	0.0111 *
Female 12–16 vs. >16	0.9388	0.2940	0.8507	0.2543
Male				
ANOVA	<0.0001 *	<0.0001 *	<0.0001 *	0.0001 *
Tukey’s				
Male 4–8 vs. 8–12	0.9949	0.9957	0.3726	0.1813
Male 4–8 vs. 12–16	0.9668	0.3701	0.3675	0.0592
Male 4–8 vs. >16	<0.0001 *	<0.0001 *	0.0003 *	0.0002 *
Male 8–12 vs. 12–16	0.9876	0.2599	0.9890	0.7343
Male 8–12 vs. >16	<0.0001 *	<0.0001 *	0.0012 *	0.0071 *
Male 12–16 vs. >16	0.0001 *	0.0008 *	0.0692	0.5206

ANOVAs and Tukey’s post hoc tests were performed. * Statistically significant (*p* < 0.05). Ratio 1: Nose width: ICW; Ratio 2: Philtrum width: ICW; Ratio 3: Mouth width: ICW. ICW: intercanthal width.

**Table 5 jcm-11-07257-t005:** Analysis results: ratios for pouting faces.

Female	Female 4–8 Years (*n* = 13)	Female 8–12 Years (*n* = 60)	Female 12–16 Years (*n* = 17)	Female >16 Years (*n* = 78)
Ratio 1: Nose width: Intercanthal width (SD)	0.90 (0.07)	0.91 (0.08)	0.94 (0.10)	0.92 (0.11)
Ratio 2: Philtrum width: Intercanthal width (SD)	0.21 (0.04)	0.23 (0.03)	0.26 (0.03)	0.23 (0.04)
Ratio 3: Mouth width: Intercanthal width (SD)	0.92 (0.15)	1.06 (0.14)	1.19 (0.13)	1.08 (0.18)
Intercanthal width in mm (SD)	31.2 (2.56)	32.5 (2.39)	32.8 (2.55)	34.3 (3.98)
Male	Male 4–8 years (*n* = 17)	Male 8–12 years (*n* = 58)	Male 12–16 years (*n* = 22)	Male >16 years (*n* = 63)
Ratio 1: Nose width: Intercanthal width (SD)	0.94 (0.06)	0.93 (0.07)	0.94 (0.08)	0.99 (0.11)
Ratio 2: Philtrum width: Intercanthal width (SD)	0.23 (0.04)	0.24 (0.04)	0.26 (0.04)	0.25 (0.04)
Ratio 3: Mouth width: Intercanthal width (SD)	0.99 (0.13)	1.09 (0.16)	1.16 (0.14)	1.12 (0.14)
Intercanthal width in mm (SD)	30.7 (1.84)	32.2 (2.34)	33.2 (1.91)	35.0 (2.78)

SD: Standard deviation; mm: millimeters.

**Table 6 jcm-11-07257-t006:** Analysis results: Statistical analysis for age groups for pouting faces.

Women	Ratio 1 Nose: ICW	Ratio 2 Phi: ICW	Ratio 3 Mouth: ICW	ICW
ANOVA	0.5836	0.0048 *	0.0002 *	0.0012 *
Tukey’s				
Female 4–8 vs. 8–12	0.9881	0.1948	0.0357 *	0.5789
Female 4–8 vs. 12–16	0.6448	0.0030 *	<0.0001 *	0.5462
Female 4–8 vs. > 16	0.8982	0.2935	0.0056 *	0.0102 *
Female 8–12 vs. 12–16	0.6211	0.0617	0.0189 *	0.9840
Female 8–12 vs. > 16	0.9204	0.9655	0.7592	0.0081 *
Female 12–16 vs. > 16	0.8378	0.0231 *	0.0815	0.3262
Men				
ANOVA	0.0007 *	0.017 *	0.0012 *	<0.0001 *
Tukey’s				
Male 4–8 vs. 8–12	0.9946	0.5966	0.0610	0.1198
Male 4–8 vs. 12–16	0.9944	0.0699	0.0014 *	0.0104 *
Male 4–8 vs. > 16	0.0799	0.0465 *	0.0035 *	<0.0001 *
Male 8–12 vs. 12–16	0.9325	0.2770	0.1678	0.3766
Male 8–12 vs. > 16	0.0006 *	0.1748	0.4868	<0.0001 *
Male 12–16 vs. > 16	0.0960	0.9891	0.7224	0.0113 *

ANOVAs and Tukey’s post hoc tests were performed. * Statistically significant (*p* < 0.05). Ratio 1: Nose width: ICW; Ratio 2: Philtrum width: ICW; Ratio 3: Mouth width: ICW. ICW: intercanthal width.

## Data Availability

Not applicable.

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
