# Peer review of "Creating Three-Dimensional Templates of Smiling and Pouting Faces for Different Sex- and Age Groups"

_jcm, 2022, doi:10.3390/jcm11247257_

Round 1

Reviewer 1 Report

The paper is an accurate analysis of the facial features and their possible change over the years.  in this sense, the topic appears to be of high interest in the facial chrurgical field, in particular given the strong attention nowadays given to appearance and aging.  The parameters evaluated by the authors could be included within a 3D evaluation of the face in the preoperative planning phases

Author Response

We appreciate the reviewer for the time in reviewing our paper and providing comments. Since there were no specific comments added in the review report, no response will be written here. The authors have carefully considered the comments on the references, research design and presentation of results and tried our best to address them. Revisions are marked using the “Track Changes” function in MS Word. 

Reviewer 2 Report

This is an interesting study, aims to create average facial models for different sex- and age groups in two facial exercises: maximum closed smile, and pouting. In total, 328 healthy subjects were included in the study. Subjects were grouped by sex, and in age categories spanning four years each, with an adult group with all ages > 16 years. The facial models were created using coherent point drift and Procrustes algorithms in MATLAB. Principal component analysis was performed, and of 20 surgical landmarks, the in-group variation using standard deviation was calculated. Three distances were analyzed: nasal width, philtral width, and mouth width. To correct for facial size, these distances were analyzed as ratio of intercanthal width. For both smiling and pouting faces, all ratios increased with ageing. 

However, there are a few problems

1. With such a small sample size without inclusion of the subjects, how can the model be called “the average face”?Although this shortcoming is mentioned in the discussion, it is a big problem. I suggest the author to include more subjects to increase the reliability of data, and exclude the face with problems like tooth anomalies, malocclusion with class II or class III skeletal pattern...

2. Why the subjects were grouped into age categories spanning four years each? A variable rate of growth and development leading to significant individual differences between children, especially for adolescents aged 12 and 16 year. This distribution between the age groups may lead to errors in the study results.

3. To study the smile, it is reasonable that maximum closed smile were designed as the facial exercise model. However, why pouting face was chosen as the model?

Author Response

We appreciate the reviewer for the precious time in reviewing our paper and providing valuable comments. Their valuable and insightful comments led to possible improvements in the current version. The authors have carefully considered the comments and tried our best to address every one of them. 

A point-by-point response to the reviewer's comments and concerns is provided below.

  1. With such a small sample size without inclusion of the subjects, how can the model be called “the average face”? Although this shortcoming is mentioned in the discussion, it is a big problem. I suggest the author to include more subjects to increase the reliability of data, and exclude the face with problems like tooth anomalies, malocclusion with class II or class III skeletal pattern...

The small sample size is, indeed, the biggest problem of our study. Patient inclusion is already closed, and the problem was that the inclusion over the ages was asymmetrical due to the location of the inclusion i.e. the university museum, which was mainly visited by youngsters accompanied by parents and/or grandparents. Still, we believe that the results are very useful. Our suggestion is that future studies would include a larger population, since it would be very unpractical to restart the study and include more subjects. We further discussed the shortcoming in the discussion (line 360-368).

  1. Why the subjects were grouped into age categories spanning four years each? A variable rate of growth and development leading to significant individual differences between children, especially for adolescents aged 12 and 16 year. This distribution between the age groups may lead to errors in the study results.

After collection of data, subjects were grouped in age categories spanning four years. Retrospectively, this was the ideal group size with enough subjects for analysis power, without combining too heterogenic subjects in the same group. We added this in the discussion (line 362-364).

For the statistical analysis, it is true that unequal group sizes might affect the ANOVA. The main issue would be that unequal sample sizes affect the robustness of the equal variance assumption. But unequal sample sizes are not a problem when the variances are equal. SDs were equal, and therefore the ANOVA was not affected by the unequal sample sizes. Also, to obtain more statistical power, the post-hoc test of Tukey was performed.

  1. To study the smile, it is reasonable that maximum closed smile were designed as the facial exercise model. However, why pouting face was chosen as the model?

The magnitude of specific facial movements says something about the ability to animate. For that reason, two reproducible extreme positions of the oral soft tissues were chosen as subject of the study: smiling and pouting. This is also mentioned in the introduction (line 48-50). We added a paragraph further explaining this choice in the discussion

(line 351-359: There is a difference between the type of facial expressions of the present study and most facial grading systems. Most facial grading systems use the open maximum smile. For the present study, however, the maximum closed smile and pouting, both extreme positions of the mouth, were chosen as referential facial expressions. The main objective of this study was to provide reproducible reference material, which gives an indication of perioral muscular activity. Several studies concluded that the maximum closed smile (posed smile), and pouting are the most reproducible facial exercises. Consequently, the open maximum smile, although it often has a wider range of motion, was not used in the present study.”)